# Confidence and Dispersity Speak: Characterising Prediction Matrix for Unsupervised Accuracy Estimation

## Abstract

This work focuses on estimating how well a model performs on out-of-distribution (OOD) datasets without using labels. While recent methods study the prediction confidence, this work newly reports prediction dispersity is another informative cue. Confidence reflects whether the *individual* prediction is certain; dispersity indicates how the *overall* predictions are distributed across all categories. Our key insight is that a well-performing model should give predictions with high confidence and high dispersity. Specifically, we need to consider the two properties so as to make more accurate estimates. To this end, we use the nuclear norm which has been shown to characterize both properties. In our experiments, we extensively validate the effectiveness of nuclear norm for various models (*e.g.*, ViT and ConvNeXt), different datasets (*e.g.*, ImageNet and CUB-200), and diverse types of distribution shifts (*e.g.*, style shift and reproduction shift). We show that nuclear norm is more accurate and robust in predicting OOD accuracy than existing methods. Furthermore, we validate the feasibility of other measurements (*e.g.*, mutual information maximization) for characterizing dispersity and confidence. Lastly, we study the limitation of the nuclear norm and discuss potential directions.

## 1 Introduction

Model evaluation is critical in both machine learning research and practice. The standard evaluation protocol is to evaluate a model on a held-out test set that is 1) fully labeled and 2) drawn from the same distribution as the training set. However, this way of evaluation is often infeasible for real-world deployment, where the test environments undergo distribution shifts and ground truths are not provided. In presence of a distribution shift, in-distribution accuracy may only be a weak predictor of model performance (Deng & Zheng, 2021; Garg et al., 2022). Moreover, annotating data itself is a laborious task, let alone it is impractical to label every new test distribution. Hence, a way to predict a classifier accuracy using unlabelled test data only has recently received much attention (Chuang et al., 2020; Deng & Zheng, 2021; Guillory et al., 2021; Garg et al., 2022).

In the task of accuracy estimation, existing methods typically derive model-based distribution statistics of test sets (Deng & Zheng, 2021; Guillory et al., 2021; Deng et al., 2021; Garg et al., 2022; Baek et al., 2022). Recent works develop methods based on prediction matrix on unlabeled data (Guillory et al., 2021; Garg et al., 2022). They focus on the overall confidence of the prediction matrix. Confidence refers to whether the model gives a confident prediction on an individual test data. It can be measured by entropy or maximum softmax probability. Guillory et al. (2021) show that the average of maximum softmax scores on a test set is useful for accuracy estimation. Garg et al. (2022) predict accuracy as the fraction of test data with maximum softmax scores above a threshold.

In this work, we newly consider another property of prediction matrix: dispersity. It measures how spread out the predictions are across classes. When testing a source-trained classifier on a target (out-of-distribution) dataset, target features may exhibit degenerate structures due to the distribution shift, where many target features are distributed in a few clusters . As a result, their corresponding class predictions would also be degenerate rather than diverse: the classifier predicts test features into specific classes and few into others. There are existing works that encourages the cluster sizes in the target data to be balanced (Shi & Sha, 2012; Liang et al., 2020; Yang et al., 2021; Tang et al., 2020),

thereby increasing the prediction dispersity. In contrast, this work does not aim to improve cluster structures and instead studies the prediction dispersity to predict model performance on various test sets without ground truths.

To illustrate that dispersity is useful for accuracy estimation, we report our empirical observation in Fig. 1. We compute the predicted dispersity score by measuring whether the frequency of predicted class is uniform. Specifically, we use entropy to quantify the frequency distribution, with higher scores indicating that the overall predictions are well-balanced. We show that the dispersity score exhibits a very strong correlation (Spearman's rank correlation $\rho > 0.950$) with classifier performance when testing on various test sets. This implies that when the classifier does not generalize well on the test set, it tends to give *degenerate* predictions (*i.e.*, low prediction dispersity), where the test samples are mainly assigned to some specific classes.

Based on the above observation, we propose to use nuclear norm, known to be effective in measuring both prediction dispersity and confidence (Cui et al., 2020; 2021), towards accurate estimation. Other measurements can also be used, such as mutual information maximizing (Bridle et al., 1991; Krause et al., 2010; Shi & Sha, 2012). Across various model architectures on on a range of datasets, we show that the nuclear norm is more effective than state-of-the-art methods (*e.g.*, ATC (Garg et al., 2022) and DoC (Guillory et al., 2021)) in predicting OOD performance. Using uncontrollable and severe synthetic corruptions, we show that nuclear norm is again superior. Finally, we demonstrate that the nuclear norm still makes reasonably accurate estimations for test sets with moderate imbalances of classes. We additionally discuss potential solutions under strong label shifts.

## 2 RELATED WORK

**Unsupervised accuracy estimation** is proposed to evaluate a model on unlabeled datasets. Recent methods typically consider the characteristics of unlabeled test sets (Deng & Zheng, 2021; Guillory et al., 2021; Deng et al., 2021; Garg et al., 2022; Baek et al., 2022; Yu et al., 2022; Chen et al., 2021b;a). For example, Deng & Zheng (2021); Yu et al. (2022); Chuang et al. (2020) consider the distribution discrepancy for accuracy estimation. Chen et al. (2021b) achieve more accurate estimation by using specified slicing functions in the importance weighting. Chuang et al. (2020) learn a domain-invariant classifiers on unlabeled test set to estimate the target accuracy. Guillory et al. (2021); Garg et al. (2022) propose to predict accuracy based the softmax scores on unlabeled data. In addition, agreement score of multiple models' predictions on test data is investigated in (Madani et al., 2004; Platanios et al., 2016; 2017; Donmez et al., 2010; Chen et al., 2021a). This work also focuses on estimating a model's OOD accuracy on various datasets and proposes to achieve robust estimations by considering the both prediction confidence and dispersity.

**Predicting ID generalization gap.** To predict the performance gap between *a certain pair* of training-testing set, several works explore develop complexity measurements on trained models and training data (Eilertsen et al., 2020; Unterthiner et al., 2020; Arora et al., 2018; Corneanu et al., 2020; Jiang et al., 2019a; Neyshabur et al., 2017; Jiang et al., 2019b; Schiff et al., 2021). For example, Corneanu et al. (2020) predict the generalization gap by using persistent topology measures. Jiang et al. (2019a) develop a measurement of layer-wise margin distributions for the generalization prediction. Neyshabur et al. (2017) use the product of norms of the weights across multiple layers. Baldock et al. (2021) introduce a measure of example difficulty (*i.e.*, prediction depth) to study the learning of deep models. Chuang et al. (2021) develop margin-based generalization bounds with optimal transport. The above works assume that the training and test sets are from the same distribution and they do not consider the characteristics of the test distribution. In comparison, we focus on predicting a model's accuracy on *various* OOD datasets.

**Calibration** aims to make the probability obtained by the model reflect the true correctness likelihood (Guo et al., 2017; Minderer et al., 2021). To achieve this, several methods have been developed to improve the calibration of their predictive uncertainty, both during training (Karandikar et al., 2021; Krishnan & Tickoo, 2020) and after (Guo et al., 2017; Gupta et al., 2021) training. For a perfectly calibrated model, the average confidence over a distribution corresponds to its accuracy over this distribution. However, calibration methods seldom exhibit desired calibration performance under distribution shifts (Ovadia et al., 2019; Gong et al., 2021). To estimate OOD accuracy, this work does not focus on calibrating confidence. Instead, we use the dispersity and confidence of prediction matrix to predict model performance on unlabeled data.

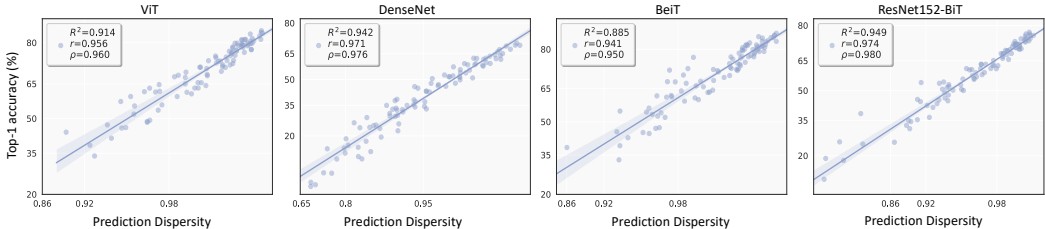

Figure 1: **Strong correlation between prediction dispersity and classifier accuracy.** Each point corresponds to one test set of ImageNet-C. The straight lines are calculated by linear regression. We study four ImageNet models (ViT, DenseNet, BeiT, and ResNet152-BiT). We compute the predicted dispersity score by measuring how uniform the frequency of predicted class is. We observe that prediction dispersity exhibits a strong correlation (Spearman's rank correlation $\rho > 0.950$) with classification accuracy for various test datasets. This indicates that if a classier gives class predictions with high dispersity, it likely achieves high accuracy, and not otherwise.

## 3 METHODOLOGY

### 3.1 PROBLEM DEFINITION

**Notations.** Consider a classification task with input space $\mathcal{X} \subseteq \mathbb{R}^d$ and label space $\mathcal{Y} = \{1, \ldots, k\}$. Let $p_S$ and $p_T$ denote source and target distributions over $\mathcal{X} \times \mathcal{Y}$, respectively. Given a source training dataset $\mathcal{D}_{\text{train}}^S$ drawn from $p_S$, we train a probabilistic predictor $f : \mathbb{R}^d \to \Delta_k$, where $\Delta_k$ denotes the $k - 1$ dimensional unit simplex. We assume a held-out test set $\mathcal{D}_{\text{test}}^S = \{(\boldsymbol{x}_i^s, y_i^s)\}_{i=1}^{n_s}$ contains $n_s$ data i.i.d sampled from $p_S$. When queried at source data $(\boldsymbol{x}^s, y^s)$ of $\mathcal{D}_{\text{test}}^S$, $f$ returns $\hat{y} =: \arg\max_{j \in \mathcal{Y}} f_j(\boldsymbol{x}^s)$ as the predicted label and $\hat{p} =: \max_{j \in \mathcal{Y}} f_j(\boldsymbol{x}^s)$ as the associated softmax confidence score. With label, we can easily compute the classification error on that data by $\mathcal{E}(f(\boldsymbol{x}^s), y^s) := \mathbf{1}_{\text{condition}}(y^s \neq \hat{y})$. By calculating the errors on all data of $\mathcal{D}_{\text{test}}^S$, we evaluate the accuracy $f$ on the source (in-distribution) $p_S$.

**Unsupervised Accuracy Estimation.** Due to distribution shift ($p_S \neq p_T$), the accuracy on in-distribution $\mathcal{D}_{\text{test}}^S$ is usually a weak estimate of how well $f$ performs on the target (out-of-distribution) $p_T$. This work aims to assess the generalization of $f$ on target (out-of-distribution) $p_T$ *without access to labels*. Concretely, given a source-trained $f$ and an unlabeled dataset $\mathcal{D}_{\text{u}}^T = \{(\boldsymbol{x}_i^t)\}_{i=1}^{n_t}$ with $n_t$ samples drawn i.i.d. from $p_T$, we aim to develop a quantity that strongly correlates with the accuracy of $f$ on $\mathcal{D}_{\text{u}}^T$. Note that, the target distribution $p_T$ has the same $K$ classes as the source distribution $p_S$ in this work (known as the closed-set setting). Unlike domain adaptation, which aims to adapt the model to the target data, unsupervised accuracy estimation focuses on predicting model accuracy on various unlabeled test sets.

### 3.2 PREDICTION CONFIDENCE AND DISPERSITY

Let $\boldsymbol{P} \in \mathbb{R}^{n_t \times k}$ denote the prediction matrix of $f$ on $\mathcal{D}_{\text{u}}^T$, and its each row $\boldsymbol{P}_{i,:}$ is the softmax vector of $i$-th target data. The values of $\boldsymbol{P}$ are in the interval $[0, 1]$. Based on the predicted class of each softmax vector, we divide $\boldsymbol{P}$ into $k$ class groups ($k$ is the number of classes). Then, we analyze the following two properties of $\boldsymbol{P}$.

**Confidence.** It measures whether a softmax vector (each row of $\boldsymbol{P}$) is certain. Common ways to measure the confidence include entropy and maximum softmax score. If the overall confidence of $\boldsymbol{P}$ is high, then it implies that the classifier $f$ is certain on the given test set. Prediction confidence has been reported to be useful in predicting classifier performance on various test sets (Guillory et al., 2021; Garg et al., 2022). For example, the overall confidence of $\boldsymbol{P}$ measured by the average of maximum softmax score is predictive of classifier accuracy (Guillory et al., 2021). Other measures such as entropy (Guillory et al., 2021; Garg et al., 2022) also give similar observation.

**Dispersity.** It is another property of $\boldsymbol{P}$ that measures whether the predicted classes are diverse and well-distributed. High dispersity means that predictions on test samples are well-distributed among $k$ classes. When testing source-train classifier $f$ on a target dataset $\mathcal{D}_{\text{u}}^T$, the target features may exhibit degenerate structures due to distribution shift. A commonly seen pattern is that many

target features are distributed in few clusters . This likely leads to degenerate predictions: the classifier tends to predict test features into some particular classes (and neglects other classes). Recent methods (Tang et al., 2020; Liang et al., 2020; Yang et al., 2022) report that regularizing prediction dispersity by encouraging cluster size to be balanced is beneficial when training domain adaptive models. Here, we study whether prediction dispersity is useful for the problem of accuracy estimation, instead of adapting models to the target domain.

To verify the usefulness of dispersity in accuracy prediction, we conduct preliminary correlation study using ImageNet-C in Fig. 1. Here, the prediction dispersity score is simply computed by measuring whether the number of softmax vectors in each class is similar: we first calculate the histogram of the sizes of predicted class, and then use entropy to measure the degree of balance. With four ImageNet models, we observe that prediction dispersity has a consistently strong correlation (rank correlation $\rho > 0.950$) with model accuracy on various test sets (ImageNet-C). This shows that when classifier does not generalize well on test data, it tends to give *degenerate* predictions (low prediction dispersity), where the test samples are mainly assigned to some specific categories.

### 3.3 CHARACTERIZING DISPERSITY AND CONFIDENCE WITH NUCLEAR NORM

Based on the above observation, we aim to quantify dispersity and confidence of prediction matrix $\boldsymbol{P}$ to estimate classification accuracy. For this purpose, we resort to the nuclear norm which is known to be effective in measuring both prediction dispersity and confidence (Cui et al., 2020; 2021).

Nuclear norm $||\boldsymbol{P}||_*$ is defined as the sum of singular values of $\boldsymbol{P}$. It is the tightest convex envelop of rank function within the unit ball (Fazel, 2002). A larger nuclear norm implies more classes are predicted and involved, indicating higher prediction dispersity. In addition, the nuclear-norm $||\boldsymbol{P}||_*$ is an upperbound of the Frobenius-norm that $||\boldsymbol{P}||_F$ reflects prediction confidence Cui et al. (2020). In Section A of the appendix, we briefly introduce how nuclear norm reflects the prediction confidence and dispersity. Since test sets can contain any numbers of data points, we normalize the nuclear norm of prediction matrix by its upper bound derived from matrix size and obtain $\widehat{||\boldsymbol{P}||_*} = ||\boldsymbol{P}||_* / \sqrt{\min(n_t, k) \cdot n_t}$. We mainly use $\widehat{||\boldsymbol{P}||_*}$ to measure the confidence and dispersity of prediction matrix $\boldsymbol{P}$ in this work. In the experiment, we also show that another measure mutual information maximization (Bridle et al., 1991; Krause et al., 2010; Shi & Sha, 2012; Yang et al., 2022) is also feasible for the task of accuracy estimation.

## 4 EXPERIMENT

### 4.1 EXPERIMENTAL SETUPS

**ImageNet-1K.** (i) Model. We use 6 representative neural networks provided by (Wightman, 2019). First, we include three vision transformers: ViT-Base-P16 (ViT) (Dosovitskiy et al., 2020), BEiT-Base-P16 (BEiT) (Liu et al., 2022), and Swin-Small-P16 (Swin) (Liu et al., 2021). Second, we include three convolution neural networks: DenseNet-121 (DenseNet), ResNetv2-152-BiT-M (Res152-BiT) (Kolesnikov et al., 2020), ConvNeXt-Base (Liu et al., 2022). They are either trained or fine-tuned on ImageNet training set (Deng et al., 2009).
(ii) Synthetic Shift. We use ImageNet-C benchmark (Hendrycks & Dietterich, 2019) to study the synthetic distribution shift. ImageNet-C is controllable in terms of both type and intensity of corruption. It contains 95 datasets that are generated by applying 19 types of corruptions (*e.g.*, blur and contrast) to ImageNet validation set. Each type has 5 intensity levels.
(iii) Real-world Shift. We consider four natural dataset shifts, including 1) dataset reproduction shift in ImageNet-V2-A/B/C (Recht et al., 2019), 2) sketch shift in ImageNet-S(ketch) (Wang et al., 2019), 3) style shift in ImageNet-R(endition) (Hendrycks et al., 2021), and 4) bias-controlled dataset shift in ObjectNet (Barbu et al., 2019). Note that, ImageNet-R and ObjectNet only shares common 113 and 200 classes with ImageNet, respectively. Following (Hendrycks et al., 2021), we sub-select the model logits for the common classes of both test sets.

**CIFAR-10** (i) Model. We use ResNet-20 (He et al., 2016), RepVGG-A0 (Ding et al., 2021), and VGG-11 (Simonyan & Zisserman, 2014). They are trained on CIFAR-10 training set.
(ii) Synthetic Shift. Similar to ImageNet-C, we use CIFAR-10-C (Hendrycks & Dietterich, 2019) to study the synthetic shift. It contains 19 types of corruptions where there are 5 intensity levels for

each type. (iii) Real-world Shift. We include three test sets: 1) CIFAR-10.1 with reproduction shift (Recht et al., 2018), 2) CIFAR-10.2 with reproduction shift (Recht et al., 2018), and 3) CINIC-10 test set that is sampled from a different database ImageNet.

**CUB-200.** We also consider fine-grained categorization with large intra-class variations and small inter-class variations (Wei et al., 2021). We build up a setup based on CUB-200-2011 (Wah et al., 2011) that contains 200 birds categories. (i) Model. We use 3 classifiers: ResNet-50, ResNet-101, and PMG (Du et al., 2020). They are pretrained on ImageNet and finetuned on CUB-200-2011 training set. We use the publicly available codes provided by (Du et al., 2020). (ii) Synthetic Shift. Following the protocol in ImageNet-C, we create CUB-200-C by applying 19 types of corruptions with 5 intensity levels to CUB-200-2011 test set. (iii) Real-world Shift. We use CUB-200-P(aintings) with style shift (Wang et al., 2020). It contains bird paintings with various rendition (*e.g.,* watercolors, oil paintings, pencil drawings, stamps, and cartoons) collected from web.

## 4.2 COMPARED METHODS AND EVALUATION METRICS

We use **four** existing measures for comparison. They are all developed based on the softmax output of classifier. **1)** *Average Confidence (AC)* (Hendrycks & Gimpel, 2017). The average of maximum softmax scores on the target dataset; **2)** *Average Negative Entropy (ANE)* (Guillory et al., 2021). The average of negative entropy scores on the target dataset; **3)** *Average Thresholded Confidence (ATC)* (Garg et al., 2022). This method first identifies a threshold on source validation set. Then, ATC is defined as the expected number of target images that obtain a softmax confidence score than the threshold; **4)** *Difference of Confidence (DOC)* (Guillory et al., 2021). It is defined as the source validation accuracy minus the difference of AC on the target dataset and source validation set. The difference of AC is regarded as a surrogate of distribution shift.

**Evaluation Procedure.** Given a trained classifier, we test it on 95 synthesized test sets under each setup. For each test set, we calculate the ground-truth accuracy and the estimated OOD quantity. Then, we evaluate the correlation strength between the estimated OOD quantity and accuracy. We also show scatter plots and mark real-world datasets to compare different approaches.

**Evaluation Metrics.** To measure the quality of estimations, we use Pearson Correlation coefficient ($r$) (Benesty et al., 2009) and Spearman's Rank Correlation coefficient ($\rho$) (Kendall, 1948) to quantify the linearity and monotonicity. They range from $[-1, 1]$. A value closer to 1 (or $-1$) indicates strong positive (or negative) correlation, and 0 implies no correlation Benesty et al. (2009). To precisely show the correlation, we use prob axis scaling that maps the range of both accuracy and estimated OOD quantity from $[0, 1]$ to $[-\infty, +\infty]$, following Taori et al. (2020); Miller et al. (2021). We also report the coefficient of determination ($R^2$) (Nagelkerke et al., 1991) of the linear fit between estimated OOD quantity and accuracy following (Yu et al., 2022). The coefficient $R^2$ ranges from 0 to 1. An $R^2$ of 1 indicates that regression predictions perfectly fit OOD accuracy.

## 4.3 MAIN RESULTS

**Nuclear norm is an effective indicator to OOD accuracy.** In Table 1, we report the correlation results of nuclear norm under three setups: ImageNet-1k, CIFAR-10, and CUB-200. We consistently observe a very strong correlation ($R^2 > 0.945$ and $\rho > 0.960$) between the nuclear norm and ODD accuracy under the three setups. The strong correlation still exists when using different model architectures under each setup. For example, the average coefficients of determination $R^2$ achieved by nuclear norm are 0.979, 0.990, and 0.989 on ImageNet-1k, CIAFR-10, and CUB-200, respectively. It demonstrates that nuclear norm well captures the distribution shift and makes excellent OOD accuracy estimations for different classifiers.

**Nuclear norm is generally more robust and accurate than existing methods.** Compared with existing methods, nuclear norm achieves the strongest correlation with classifier performance across all the three setups by considering prediction dispersity as well as the confidence. With different models on ImageNet, nuclear norm achieves an average $R^2$ of 0.979, while the second best method (ATC) only obtains 0.924. Moreover, nuclear norm outperforms ATC by 0.262 and in average $R^2$ under CUB-200 setup. Moreover, the prediction performance of nuclear norm is overall more robust than other methods. Specifically, we observe that the competing methods are less effective in predicting the accuracy of certain classifiers such as Swin under the ImageNet setup and ResNet-101

Table 1: **Method comparison under ImageNet, CIFAR-10, and CUB-200 setups**. We compare nuclear norm with four existing methods. To quantify the effectiveness in assessing OOD generalization, we report coefficients of determination ($R^2$) and Spearman's rank correlation ($\rho$). The highest score in each row is highlighted in **bold**. We show that nuclear norm exhibits the highest correlation strength ($R^2$ and $\rho$) with OOD accuracy across three setups.

| Setup | Model | AC | | ANE | | ATC | | DoC | | Nuclear Norm | |
|---|---|---|---|---|---|---|---|---|---|---|---|
| | | $R^2$ | $\rho$ | $R^2$ | $\rho$ | $R^2$ | $\rho$ | $R^2$ | $\rho$ | $R^2$ | $\rho$ |
| ImageNet | ViT | 0.970 | 0.990 | 0.964 | 0.988 | 0.978 | 0.990 | 0.961 | 0.990 | **0.991** | **0.995** |
| | BeiT | 0.977 | 0.994 | 0.964 | 0.989 | 0.985 | 0.995 | 0.979 | 0.994 | **0.988** | **0.996** |
| | Swin | 0.794 | 0.929 | 0.732 | 0.909 | 0.815 | 0.935 | 0.791 | 0.929 | **0.949** | **0.961** |
| | DenseNet | 0.938 | 0.984 | 0.929 | 0.979 | 0.961 | 0.989 | 0.937 | 0.984 | **0.995** | **0.997** |
| | Res152-BiT | 0.891 | 0.981 | 0.877 | 0.979 | 0.916 | 0.982 | 0.908 | 0.981 | **0.981** | **0.991** |
| | ConvNeXt | 0.894 | 0.971 | 0.866 | 0.960 | 0.888 | 0.967 | 0.899 | 0.971 | **0.967** | **0.982** |
| | Average | 0.911 | 0.975 | 0.889 | 0.968 | 0.924 | 0.976 | 0.911 | 0.975 | **0.979** | **0.989** |
| CIFAR-10 | ResNet-20 | 0.916 | 0.991 | 0.916 | 0.991 | 0.934 | 0.992 | 0.937 | 0.991 | **0.989** | **0.995** |
| | RepVGG-A0 | 0.811 | 0.982 | 0.806 | 0.981 | 0.841 | 0.985 | 0.824 | 0.982 | **0.992** | **0.996** |
| | VGG-11 | 0.973 | 0.994 | 0.973 | 0.995 | 0.984 | **0.996** | 0.964 | 0.994 | **0.988** | **0.996** |
| | Average | 0.900 | 0.989 | 0.900 | 0.988 | 0.920 | 0.991 | 0.908 | 0.989 | **0.990** | **0.995** |
| CUB-200 | ResNet-50 | 0.836 | 0.942 | 0.839 | 0.939 | 0.855 | 0.957 | 0.818 | 0.942 | **0.989** | **0.997** |
| | ResNet-101 | 0.303 | 0.734 | 0.319 | 0.739 | 0.351 | 0.775 | 0.308 | 0.734 | **0.987** | **0.998** |
| | PMG | 0.892 | 0.979 | 0.893 | 0.977 | 0.977 | 0.991 | 0.903 | 0.979 | **0.990** | **0.998** |
| | Average | 0.677 | 0.885 | 0.684 | 0.885 | 0.727 | 0.908 | 0.677 | 0.885 | **0.989** | **0.997** |

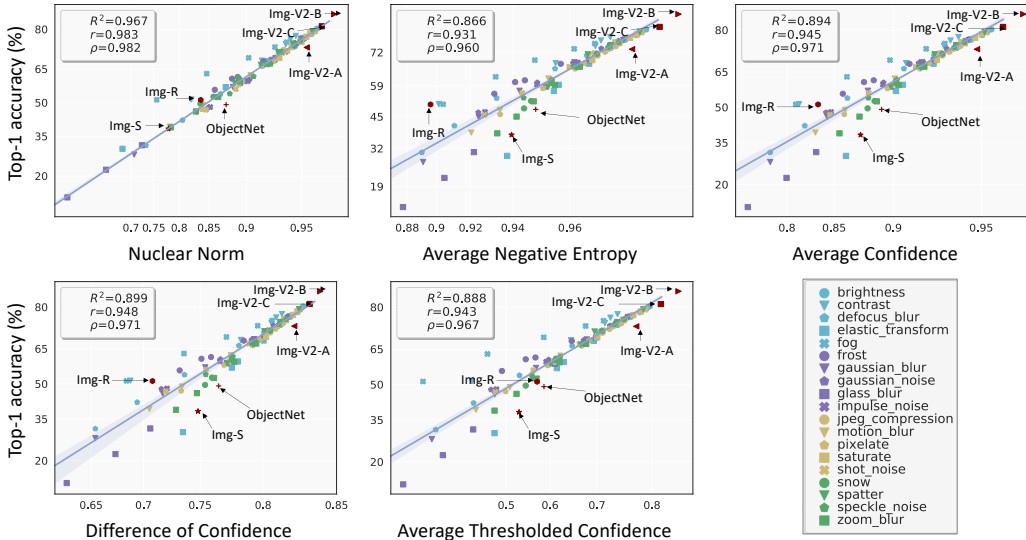

Figure 2: **Correlation study under the ImageNet setup.** We plot the actual accuracy of *ConvNeXt* and five measures including nuclear norm and four competing methods. Different shapes in each sub-figure represents different test sets. The straight lines are calculated by linear regression fit on synthetic datasets of ImageNet-C. We list the 19 types of corruptions in ImageNet-C using different shapes and colors in the bottom right figure. We also mark the 6 real-world datasets in each sub-figure with arrows. We observe nuclear norm exhibits stronger correlation with accuracy. Moreover, with nuclear norm, real-world test sets are closely around the linearly fit line.

under the CUB-200 setup. For these difficult cases, nuclear norm remains useful with $R^2 > 0.945$, which further validates its effectiveness.

**Nuclear norm can estimate accuracy of real-world datasets**. To further validate the effectiveness of nuclear norm, we show its accuracy prediction on real-world datasets as the scatter plots under

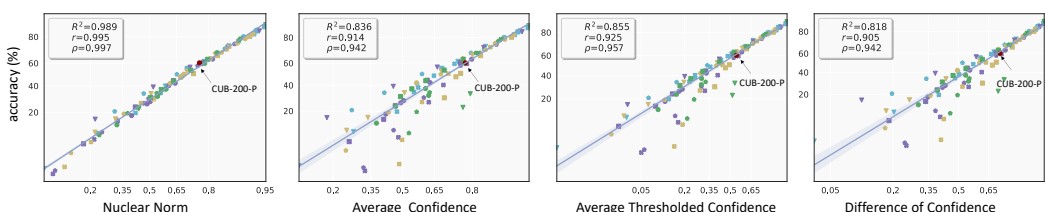

Figure 3: **Correlation study under the CIFAR-10 setup.** We plot the actual accuracy of *ResNet-20* and the estimated OOD quantity. We show the results of nuclear norm, AC and ATC. The lines are calculated by linear regression fit on CIFAR-C. We mark the 3 real-world test sets in each sub-figure. We show that AC and ATC fail to estimate generalization on datasets with lower ground-truth accuracy. In comparison, nuclear norm is more robust and accurate.

Figure 4: **Correlation study under the CUB-200 setup.** We plot the actual accuracy of *ResNet-50* and the estimated OOD quantity. We compare nuclear norm with AC and ATC. The straight lines are calculated by the linear regression fit on CUB-200-C. We mark the real-world test set CUB-P in each sub-figure. We show that AT and ATC cannot give accurate estimates for some datasets. In contrast, nuclear norm is more robust and accurate in predicting generalization: all test sets are closely around the linear line, yielding higher correlation strength.

the three setups (Fig. 2, Fig. 3, and Fig. 4, respectively). We observe that nuclear norm can produce reasonably accurate estimation on real-world test sets. Under the ImageNet setup (Fig. 2), the six test sets (*e.g.*, ImageNet-V2/A/B/C and ImageNet-R) are very close to the linear regression line. It demonstrates that nuclear norm well captures these real-world shifts and thus estimates OOD performance very well. Under CIFAR-10 and CUB-200 setups, we have similar observations.

Although existing methods (*e.g.*, ATC) are effective on most real-world datasets, nuclear norm still shows its advantage over them. Other methods fail to capture the shifts of ImageNet-S and ObjectNet under the ImageNet setup: they are far away from linear lines. In comparison, nuclear norm captures them well and both datasets are very close to linear lines. Furthermore, the scatter plots under the CIFAR-10 (Fig. 3) and CUB-200 (Fig. 4) show that the competing methods often give accuracy numbers lower than the ground truth when the test set is difficult, while nuclear norm is still effective.

## 4.4 DISCUSSION AND ANALYSIS

**(I) Beyond controllable synthetic shifts.** The synthetic datasets (*e.g.*, ImageNet-C) are algorithmically generated in a controllable manner. Here, we investigate whether a measure is robust in predicting OOD accuracy on random synthetic datasets. To this end, we randomly synthesize datasets for the *CIFAR-10 setup*. Specifically, we use 10 new corruptions of ImageNet-$\bar{C}$ (Mintun et al., 2021) that are *perceptually dissimilar* to ImageNet-C. The dissimilar corruptions include warps, blurs, color distortions, noise additions, and obscuring effects. When synthesizing each test set, we randomly choose 3 corruptions and make corruption strength random. By doing so, we create 200 random synthetic datasets denoted CIFAR-$\bar{C}$-Rand.

In Fig. 5, we report the correlation results using ResNet-20 under the CIFAR-10 setup. We also show the linear regression lines that are fit on datasets of CIFAR-10-C. We report results of four methods including nuclear norm, AC, ATC, and DoC. We have two observations. First, for each method, CIFAR-$\bar{C}$-Rand datasets (marked with "+") are generally distributed around the linear lines. This indicates that all methods can make reasonable accuracy estimations on CIFAR-$\bar{C}$-Rand. Second,

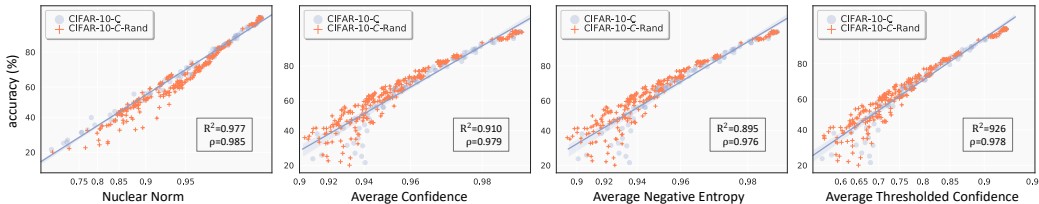

Figure 5: **Correlation study on randomly synthesized datasets under the CIFAR-10 setup.** We report results with ResNet-20. Randomly synthesized datasets (CIFAR-10-$\bar{C}$-Rand) are marked with orange "+", and the solid lines are fit with robust linear regression on controllable CIFAR-10-C. Overall, CIFAR-10-$\bar{C}$-Rand datasets are distributed around the linear line for every method. Looking more closely at low-accuracy region (bottom left in each subfigure), nuclear norm gives more accurate estimations than other methods, indicating its effectiveness.

for the low-accuracy region (bottom left in each subfigure), nuclear norm gives more accurate and robust predictions than other methods (*e.g.*, ATC and DoC).

**(II) Other measures to consider prediction confidence and dispersity.** Here, we discuss the usage of other measures. We study mutual information maximizing (MI) which is commonly used in discriminative clustering (Bridle et al., 1991; Krause et al., 2010). Recent methods use it as a regularization to make model predictions confident and diverse (Liang et al., 2020; Yang et al., 2021; Tang et al., 2020). Given a prediction matrix $P \in \mathbb{R}^{n_t \times k}$, IM is defined as $H\left(\frac{1}{n_t}\sum_{i=1}^{n_t} P_{i,:}\right) - \frac{1}{n_t}\sum_{i=1}^{n_t} H(P_{i,:})$. Its first term encourages the predictions to be globally balanced. The second term is standard entropy that makes the prediction confident. In Table 2, we report the correlation results using MI. We observe that MI and nuclear norm achieve similar average correlation strength.

| Method | ImageNet | CIFAR | CUB |
|---|---|---|---|
| ANE | 0.968 | 0.988 | 0.885 |
| MI | 0.982 | 0.994 | 0.995 |
| Nuclear Norm | 0.989 | 0.995 | 0.997 |

Table 2: **Correlation results using mutual information maximizing (MI).** We report the average correlation strength (Spearman's rank correlation $\rho$) under each setup. We observe MI and nuclear norm have similar correlation strength. Compared with average negative entropy (ANE), MI exhibits stronger correlation with accuracy across three setups.

achieve similar average correlation strength. Compared with average negative entropy (ANE), MI exhibits stronger correlation across three setups. For example, MI yields a $0.110$ higher $\rho$ than ANE on CUB. This further validates that prediction dispersity is informative for accuracy estimation.

**III. Effect of test set size.** As illustrated in Section 3.3, nuclear norm without scaling is related to the size of the prediction matrix. Since test sets can contain any numbers of data points, we normalize nuclear norm by its upper bound. Here, we change the size of each dataset of ImageNet-C by randomly selecting $20–90\%$ of all test samples. As shown in Fig. 6, scaled nuclear norm is well correlated with accuracy under different test set sizes.

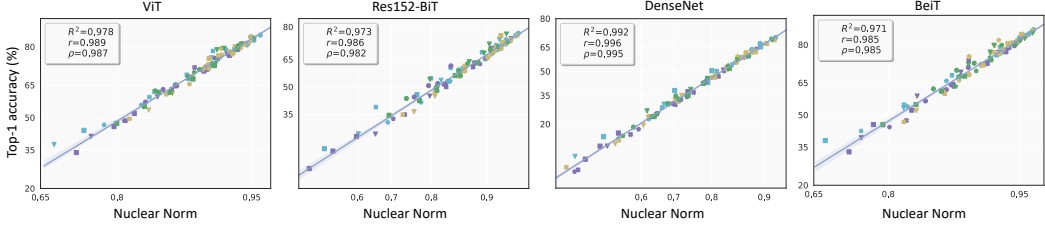

Figure 6: **Analysis of the influence of test set size on nuclear norm.** We conduct correlation study on *randomly sub-sampled* ImageNet-C. Specifically, we vary the size of each dataset by randomly selecting $20–90\%$ of test samples. We test three classifiers and observe the correlation strength remains very high ($R^2 > 0.960$ and $\rho > 0.970$).

**(IV) Discussion on label shift (class imbalance).** In our work, we consider the common covariate shift (Sugiyama & Kawanabe, 2012) where $p_S(x) \neq p_T(x)$ and $p_S(y|x) = p_T(y|x)$ (*i.e.*, the class label of the input data is independent of distribution). Nuclear norm measures the prediction dispersity and thus implicitly assumes that the test set *does not contain strong* label shift (*i.e.*,

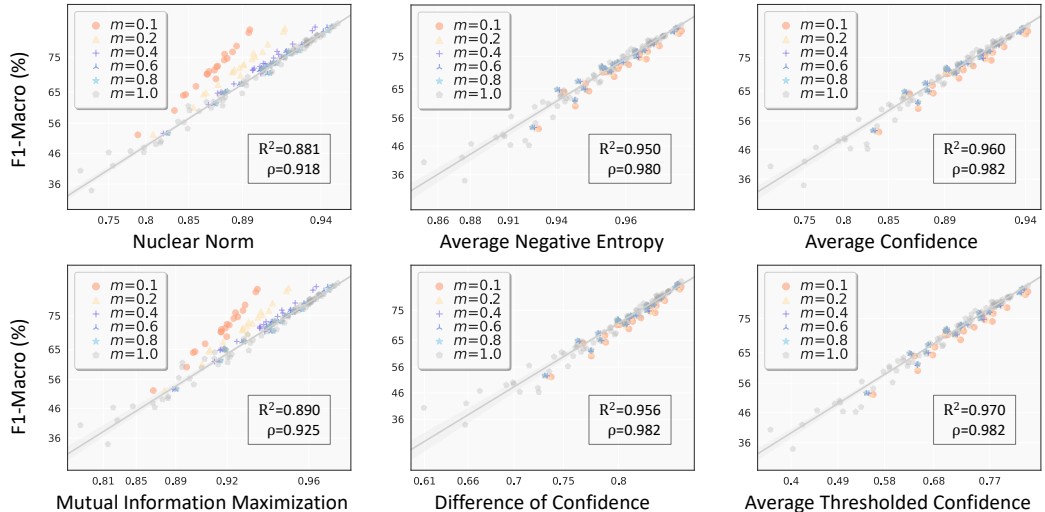

Figure 7: **Comparison of various methods on imbalanced test sets.** Using **ViT** under ImageNet setup, we study the robustness of existing methods to several imbalance ratio $m$ when test sets are long-tailed. A smaller $m$ indicates a higher imbalance intensity. The linear lines are fit on standard test sets ($m = 1$). We observe that both mutual information maximization (MI) and nuclear norm are less effective than other methods under strong-imbalanced datasets ($m < 0.4$). Furthermore, we show that MI and nuclear norm are robust under mild-imbalanced test sets ($m \geq 0.4$).

class imbalance). As for the label shift (Garg et al., 2020), the assumption about the distribution is $p_S(y) \neq p_T(y)$ and $p_S(\boldsymbol{x}|y) = p_T(\boldsymbol{x}|y)$ (*i.e.*, the class-conditional distribution does not change).

Here, we discuss the robustness of nuclear norm to label shift. We first note that real-world test sets such as ImageNet-R, ObjectNet, and CUB-200-P are already imbalanced. We show that nuclear norm robustly captures them: they are very close to the linear lines (as shown in Fig. 2 and Fig. 4) To further study the effect of label shift, we create long-tailed imbalance test sets. We use an exponential decay to make the proportion of each class different between source and target datasets following Cao et al. (2019). We use imbalance ratio $m$ to denote the ratio between sample sizes of the least frequent and most frequent class. We test several imbalanced ratios: $\{0.1, 0.2, 0.4, 0.6, 0.8\}$. We conduct the experiment on ImageNet-C and use 19 types of corruption datasets with the second intensity level. As shown in Fig. 7, we observe that both nuclear norm and MI are influenced by label shift when the imbalance is strong ($m < 0.4$). For example, when the test set is of extreme class imbalance ($m = 0.1$), the prediction of nuclear norm is not accurate. We also observe that under the strong imbalance ($m < 0.4$), exiting methods (*e.g.*, ATC) is more stable than nuclear norm and MI. We further note that nuclear norm and MI are robust to the mild imbalance ($m \geq 0.4$).

The above analysis indicates that nuclear norm is robust in the presence of moderate label shift. To deal with strong label shift, using extra techniques such as label shift estimation (Lipton et al., 2018; Tian et al., 2020) and prior knowledge (Chen et al., 2021b; Sun et al., 2022) would be helpful. It would be interesting to further study this idea in future work.

## 5 CONCLUSION

This work studies OOD accuracy estimation where the goal is to predict classifier accuracy on unlabeled test sets. While existing methods study the confidence of prediction matrix on unlabelled data, this work newly considers the prediction dispersity. It measures whether the overall predictions are well-balanced across classes. We report that prediction dispersity is a useful property which correlates strongly with classifier accuracy on various test sets. Driven by this new observation, we consider both prediction confidence and dispersity to achieve more accurate estimation. To this end, we use the nuclear norm of prediction matrix to characterize both properties. Across three setups, we consistently observe that nuclear norm is more effective and robust in assessing classifier OOD performance than existing methods. We further conduct experiment on imbalanced test sets and show that nuclear norm is still effective under moderate class imbalances.

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

# A    NUCLEAR NORM

Let $\boldsymbol{P} \in \mathbb{R}^{n_t \times k}$ denote the prediction matrix of $f$ on $\mathcal{D}_u^T$, nuclear norm $||\boldsymbol{P}||_*$ is the sum of singular values of $\boldsymbol{P}$. Nuclear norm is the tightest convex envelop of rank function within the unit ball (Fazel, 2002). A larger nuclear norm implies more classes are predicted and involved, indicating higher prediction dispersity. In addition, nuclear norm $||\boldsymbol{P}||_*$ and Frobenius norm $||\boldsymbol{P}||_F = \sqrt{Trace(\boldsymbol{P}^\intercal \boldsymbol{P})}$ can bound each other (Recht et al., 2010; Fazel, 2002). More specifically, they have the following relationship: $1/\sqrt{d}||\boldsymbol{P}||_* \leq ||\boldsymbol{P}||_F \leq ||\boldsymbol{P}||_* \leq \sqrt{d}||\boldsymbol{P}||_F$, where $d = min(n_t, k)$. In our work, because $\boldsymbol{P}$ is consists of softmax vectors, its Frobenius norm is bound by $||\boldsymbol{P}||_F \leq \sqrt{n_t}$.

Frobenius norm $||\boldsymbol{P}||_F$ reflects prediction confidence Cui et al. (2020). Based on the above relationship, a larger nuclear norm $||\boldsymbol{P}||_*$ implies a larger Frobenius norm $||\boldsymbol{P}||_F$, indicating a higher prediction confidence. Therefore, nuclear norm $||\boldsymbol{P}||_*$ can be use to characterize both confidence and dispersity of $\boldsymbol{P}$. Moreover, nuclear norm $||\boldsymbol{P}||_*$ is related to the shape of $\boldsymbol{P}$, so we normalized it by its upper bound $\sqrt{d \cdot n_t}$ and obtain $\widehat{||\boldsymbol{P}||_*} = ||\boldsymbol{P}||_* / \sqrt{d \cdot n_t}$. In our work, we use $\widehat{||\boldsymbol{P}||_*}$ to measure the confidence and dispersity of prediction matrix.

# B    EXPERIMENTAL SETUP

## B.1    MODELS

**ImageNet.** Models are provided by PyTorch Image Models (timm-1.5) Wightman (2019). They are either trained or fine-tuned on the ImageNet-1k training set Deng et al. (2009).

**CIFAR-10.** We train models using the implementations from https://github.com/chenyaofo/pytorch-cifar-models. CIFAR-$\bar{C}$-Rand is generated with the 10 new corruptions of ImageNet-$\bar{C}$ (Mintun et al., 2021) that are *perceptually dissimilar* to ImageNet-C. We apply random corruptions based on the codes from https://github.com/facebookresearch/augmentation-corruption.

**CUB-200.** We train CIFAR models using the implementations from https://github.com/PRIS-CV/PMG-Progressive-Multi-Granularity-Training. CUB-200-C is generated based on the implementations from https://github.com/hendrycks/robustness.

## B.2    DATASETS

The datasets we use are standard benchmarks, which are publicly available. We have double-checked their license. We list their open-source as follows.

**CIFAR-10** Krizhevsky et al. (2009) (https://www.cs.toronto.edu/ kriz/cifar.html);
**CIFAR-10-C** Hendrycks & Dietterich (2019) (https://github.com/hendrycks/robustness);
**CIFAR-10.1** Recht et al. (2018) (https://github.com/modestyachts/CIFAR-10.1);
**CINIC** Chrabaszcz et al. (2017) (https://github.com/BayesWatch/cinic-10).

**ImageNet-Validation** Deng et al. (2009) (https://www.image-net.org);
**ImageNet-V2-A/B/C** Recht et al. (2019) (https://github.com/modestyachts/ImageNetV2);
**ImageNet-Corruption** Hendrycks & Dietterich (2019) (https://github.com/hendrycks/robustness);
**ImageNet-Sketch** Wang et al. (2019) (https://github.com/HaohanWang/ImageNet-Sketch);
**ImageNet-Rendition** Hendrycks et al. (2021) (https://github.com/hendrycks/imagenet-r);
**ObjectNet** Barbu et al. (2019) (https://objectnet.dev).

**CUB-200-2011** Wah et al. (2011) (https://www.vision.caltech.edu/datasets/cub_200_2011). **CUB-Paintings** Wang et al. (2020) (https://github.com/thuml/PAN).

## B.3    COMPUTATION RESOURCES

We run all experiment on one 3090Ti with PyTorch (1.11.0+cu113). CPU is AMD Ryzen 9 5900X 12-Core Processor.

## B.4 EXPERIMENTAL DETAIL

**(I) Effect of temperature.** We empirically find that using a small temperature for softmax is helpful for all methods. Therefore, we use temperature of $0.4$ for all methods in the experiment. We show the effect of temperature in term of correlation strength ($R^2$ and $\rho$) in Fig. 8. We have two observations. **First**, using a small temperature (*e.g.*, $0.4$) helps for all methods including nuclear norm, ATC and DoC. The correlation results are stable when temperature ranges from $0.2$ to $0.45$. **Second**, when using various temperature values, nuclear norm consistently achieve stronger correlation.

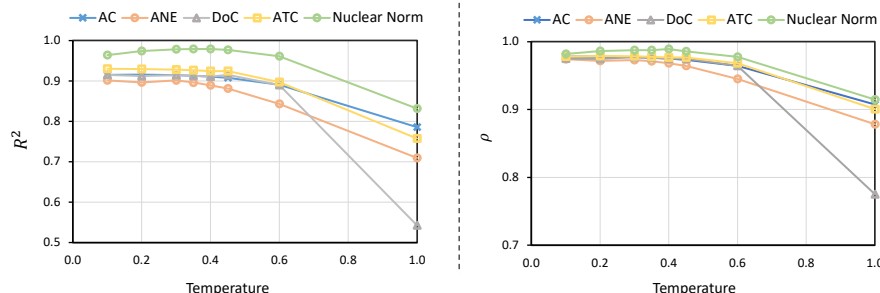

Figure 8: **Effect of temperature for all methods.** We report the correlation results (both $R^2$ and $\rho$) using various temperature of softmax. We show that a small temperature ($0.2$ to $0.45$) helps for all methods. Moreover, when using different temperature values, nuclear norm consistently exhibits a stronger correlation than other methods.

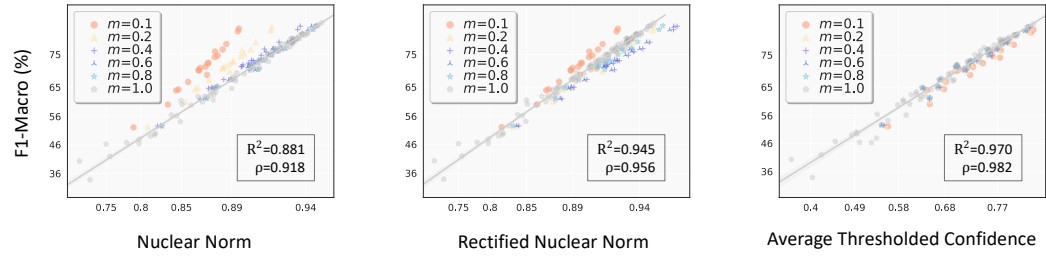

Figure 9: **Effect of rectified nuclear norm.** Under imbalanced test sets, we relax the regularization of nuclear norm on "tail" classes (rectified nuclear norm). We conduct correlation study on *imbalanced* ImageNet-C using **ViT**. We observe that rectified nuclear norm can improve the nuclear norm under imbalanced test sets.

**(III) Comparison with ALine-D (Baek et al., 2022) and ProjNorm (Yu et al., 2022).** For a fair comparison, we follow the same setting as (Baek et al., 2022) and report the results using ResNet18 on CIFAR-10-C. As shown in Table 3, we observe that nuclear norm gains stronger correlation strength than the two methods. It achieves $0.997$ and $0.990$ in rank correlation ($\rho$) and coefficients of determination ($R^2$), respectively. Furthermore, we would like to mention that ALine-D (Baek et al., 2022) requires a set of models for accuracy estima-

| Correlation | ProjNorm | ALine-D | Nuclear Norm |
|---|---|---|---|
| $\rho$ | 0.980 | 0.995 | **0.997** |
| $R^2$ | 0.973 | 0.974 | **0.990** |

Table 3: **Method comparison under CIFAR-10 setup.** We report the average correlation strength (Spearman's rank correlation $\rho$ and coefficients of determination $R^2$).

tion. ProjNorm (Yu et al., 2022) requires fine-tuning a pre-trained network on each OOD test set with pseudo-labels. In contrast, nuclear Norm is more efficient: it is computed on a classifier's prediction matrix on each unlabeled test set.

**(II) Rectified nuclear norm for imbalanced test sets**. We tried to relax the regularization of nuclear norm under imbalanced test sets. Nuclear Norm encourages the predictions to be well-distributed across all classes. For imbalanced test sets, we can relax this regularization on the tail classes. That is, we mainly consider the prediction dispersity of head classes.

To achieve this, we explored one intuitive way to rectify the nuclear norm: we modify the normalization (i.e., upper bound) of the nuclear norm. Specifically, we revise the normalization from $\sqrt{\min(n_t, k) * n_t}$ to $\sqrt{\min(n_t, k_{\text{head}}) * n_t}$, where $k_{\text{head}}$ is the number of major classes regularised by nuclear norm. We conducted experiment under ImageNet setup (k=1000) and empirically set $k_{\text{head}}$ based on the imbalanced intensity $r_m$ (the ratio between the number of last 10 "tail" classes and the number of top 10 "head" classes): $k_{\text{head}} = k - (1 - r_m) * 80$. To estimate imbalanced intensity, we use BBSE (Lipton et al., 2018) to estimate the class distribution.

In Figure 9, we show that our attempt (rectified nuclear norm) can improve nuclear norm. We would like to view the above experiment as a starting point that inspires more research on the rectification of nuclear norm for strong imbalanced test sets.

**(IV) Additional observations. First**, ObjectNet of ImageNet setup is built in a bias-controlled manner (with controls for rotation, background, and viewpoint). We observe that its images are often confidently misclassified, which makes predictions with the high nuclear norm. We believe this is why ObjectNet is always off the linear line. **Second**, for all accuracy estimation methods, they can well capture the model performance is high (top-right region of each figure). However, when model accuracy is low (bottom-left), existing methods cannot make reasonable estimations, especially under CIFAR-10 and CUB-200. In contrast, nuclear norm can well handle the low-accuracy region by additionally considering the prediction dispersity. To improve the accuracy estimation, it would be helpful to further consider the characteristics of predictions when the model performs poorly. **Third**, in Figures 2, 3, and 4, we observe that the real-world test sets (*e.g.*, ImageNet-R, CINIC, and CUB-P) scatter around the linear lines fit on synthetic datasets. This indicates that both real-world and synthetic datasets follow a similar linear trend. This gives an interesting hint: we can use synthetic datasets to simulate and capture the distributions of real-world test sets.

## B.5 MORE CORRELATION RESULTS

### B.5.1 IMAGENET SETUP

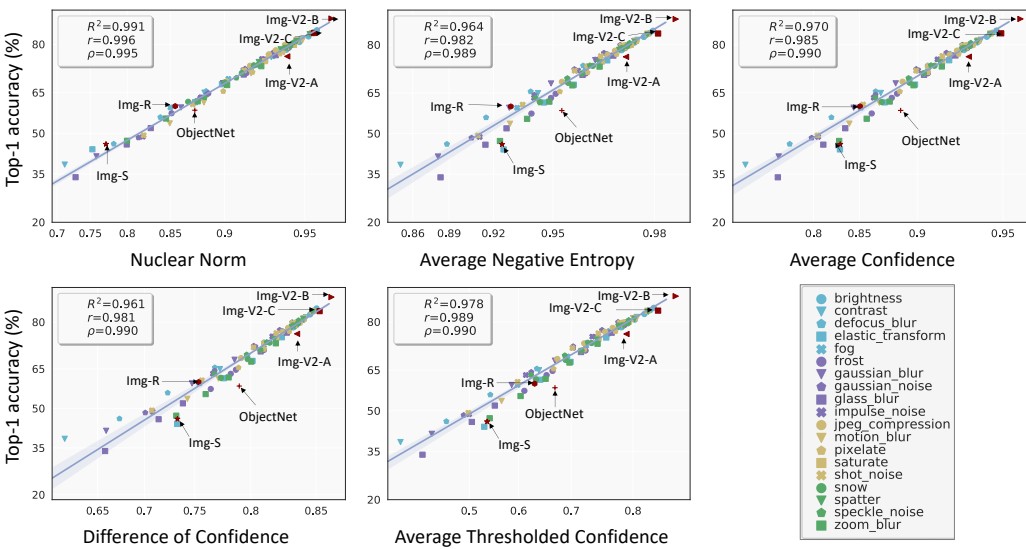

Figure 10: **Correlation study under the ImageNet setup.** We plot the actual accuracy of *ViT* and five measures including nuclear norm and four competing methods. Different shapes in each sub-figure represents different test sets. The straight lines are calculated by linear regression fit on synthetic datasets of ImageNet-C. We list the 19 types of corruptions in ImageNet-C using different shapes and colors in the bottom right figure. We also mark the 6 real-world datasets in each sub-figure with arrows. Compared with other methods, nuclear norm exhibits stronger correlation with accuracy. Moreover, with nuclear norm, real-world test sets are closely around the linearly fit line.

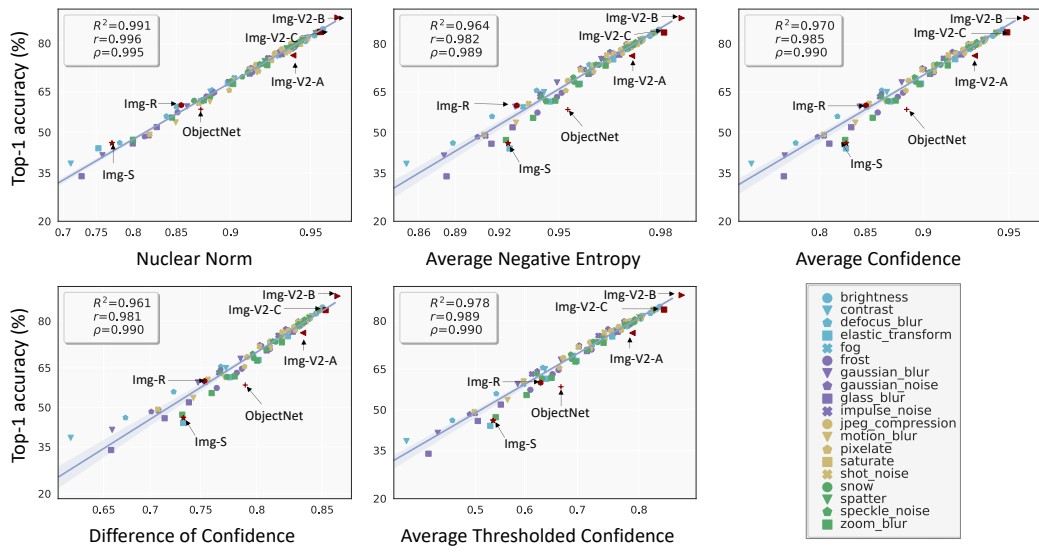

Figure 11: **Correlation study under the ImageNet setup.** We plot the actual accuracy of *BeiT* and five measures including nuclear norm and four competing methods.

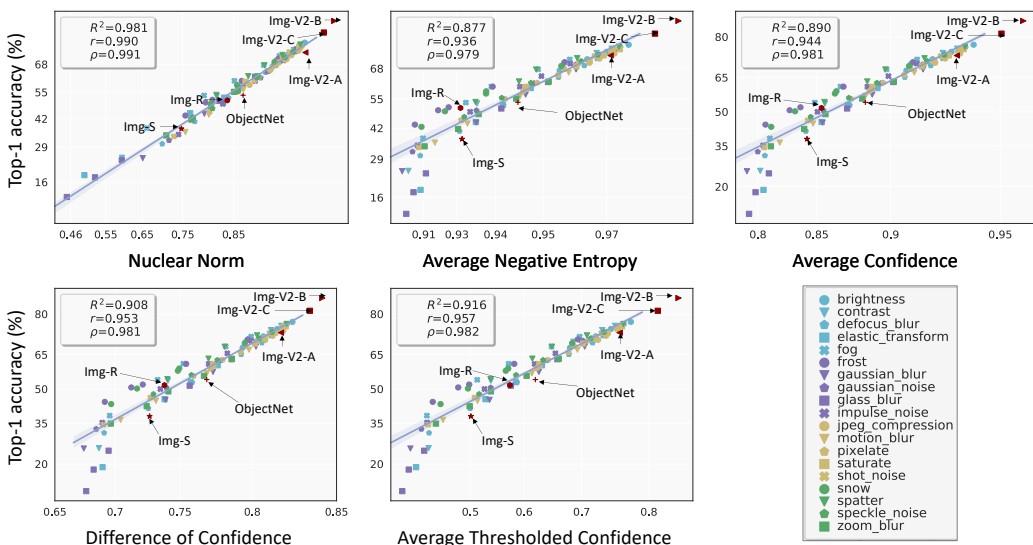

Figure 12: **Correlation study under the ImageNet setup.** We plot the actual accuracy of *Res152-BiT* and five measures including nuclear norm and four competing methods.

### B.5.2 CIFAR-10 AND CUB-200 SETUPS

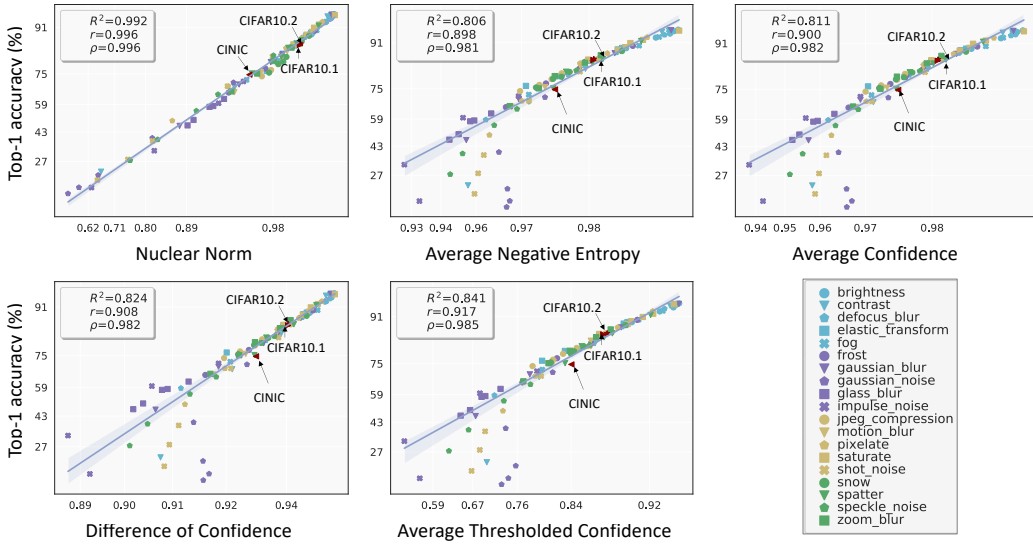

Figure 13: **Correlation study under the CIFAR-10 setup.** We plot the actual accuracy of *RepVGG-A0* and five measures including nuclear norm and four competing methods. The straight lines are calculated by linear regression fit on synthetic datasets of CIFAR-10-C. We list the 19 types of corruptions in CIFAR-10-C using different shapes and colors in the bottom right figure. We also mark the 3 real-world datasets in each sub-figure with arrows. Compared with other methods, nuclear norm exhibits stronger correlation with accuracy.

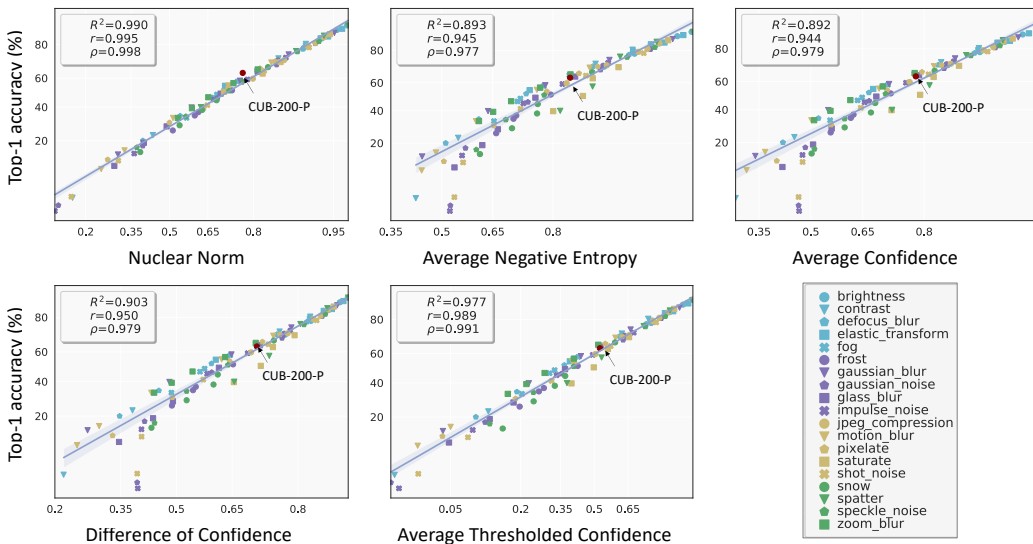

Figure 14: **Correlation study under the CUB-200 setup.** We plot the actual accuracy of *PMG* and five measures including nuclear norm and four competing methods. The straight lines are calculated by linear regression fit on synthetic datasets of CUB-200-C. We list the 19 types of corruptions in CUB-200-C using different shapes and colors in the bottom right figure. We also mark the real-world CUB-200-P in each sub-figure with arrows. Compared with other methods, nuclear norm exhibits stronger correlation with accuracy.

