# OpenReview forum: "Confidence and Dispersity Speak: Characterising Prediction Matrix for Unsupervised Accuracy Estimation"
_ICLR.cc/2023/Conference — Submitted to ICLR 2023_

### Official Review · Reviewer_JXCX · 2022-10-20

**Confidence:** 2
**Correctness:** 3
**Technical Novelty And Significance:** 3
**Empirical Novelty And Significance:** 1
**Recommendation:** 3

**Clarity, Quality, Novelty And Reproducibility:**

Some typos:
1. The second line in the Notation of Section 3.1: \Delta_k was introduced in the where clause without being used before in the sentence
2. The fourth line in the Notation of Section 3.1: it should be \Delta_k instead of \Delta_n


**Strength And Weaknesses:**

Strength: it is an interesting thought to consider the distribution of the predicted labels and use them as a metric to evaluate the generalization capability for the out-of-distribution data.

Weakness: I do have some serious concerns about the underlying assumptions used in the paper. To me the dispersity assumption may be too strong for a lot of real applications. For those image data sets used in the paper (imgnet, cifar10, CUB-200), those assumptions seem perfect since the class labels on the shifted test data are distributed relatively evenly. However for real world applications, the labels may be distributed very differently. For example some classes are rarely observed, and some classes are observed very frequently. The power law and long tail effects are prevalent. In such a situation, the dispersity assumption used in the paper is no longer valid, and thus the proposed metric may not work. This makes me wonder if the good performance shown in the paper is only an artifact of the “label-balanced” data sets.


**Summary Of The Paper:**

The authors of the paper argue that confidence and dispersity are two important factors regarding the performance on the out-of-distribution datasets. Along this line, the authors propose to use the (normalized) nuclear norm of the prediction matrix as a metric for the generalization capabilities of the models. Various experiments on image datasets show there exists a strong correlation between the accuracy on the out-of-distribution data and the proposed metric.


**Summary Of The Review:**

Overall my review for this paper tends to be negative due to the major concerns I stated in the previous comments. If the good empirical performance shown is mostly due to the artifacts of the relatively label-balanced datasets then I feel maybe it is better to introduce more new datasets that have highly unbalanced classes to test the robustness of the models instead of making use of the artifact for a little improved numbers.

---

> ### Author Response · Authors · 2022-11-16
> **Author Response to Reviewer JXCX**
>
> **Q1-1: To me the dispersity assumption may be too strong for a lot of real applications. However for real world applications, the labels may be distributed very differently. This makes me wonder if the good performance shown in the paper is only an artifact of the “label-balanced” data sets.**
> Thank you for raising this discussion. We would like to clarify the assumption of dispersity for the task of accuracy estimation as follows:
>
> Prediction dispersity measures how the predictions distribute across classes. Without additional knowledge about the class distribution, a common assumption is that the predictions are expected to be well-distributed across classes (e.g., domain adaptation (Yang et al., 2021), contrastive learning (Wang et al., 2020), and discriminative clustering (Jabi et al., 2019)). This work follows this underlying assumption, and what is more, we show that ***nuclear norm is also useful for moderately imbalanced test sets***. In the experiment (Section 4.4), we observed that:
>
> - Nuclear norm well captures existing imbalanced real-world test sets. For example, ObjectNet and ImageNet-R under ImageNet setup (Figure 2) and CUB-P under CUB-200 setup (Figure 4);
>
> - We systematically analyze the robustness of nuclear norm to various long-tailed intensities. Specifically, we tested several imbalanced ratios (m) and a smaller m denotes a higher imbalanced intensity. We show that the nuclear norm has resistance to the moderate imbalance: it still makes reasonably good estimations on moderate long-tailed test sets (m>=0.4).
>
>   *[a] Wang et al.,  "Understanding contrastive representation learning through alignment and uniformity on the hypersphere." In ICML, 2020*
>
>   *[b] Yang et al., “Exploiting the intrinsic neighborhood structure for source-free domain adaptation” In NeurIPS, 2021*
>
>   *[c] Jabi, et al. "Deep clustering: On the link between discriminative models and k-means." IEEE transactions on pattern analysis and machine intelligence, 2019*
>
>
> **Q1-2: it is better to introduce more new datasets that have highly unbalanced classes to test the robustness of the models.**
>
> Thanks for this suggestion. We agree that under highly imbalanced datasets, the underlying assumption could be too strong. To study this, we have included more analysis and discussion:
>
> - ***First***, compared with existing methods, we observe that nuclear norm is less robust under severely long-tailed test sets.  Specifically, when m<0.4 in Figure 7, the estimations of nuclear norm are less accurate than other methods.
>
> - ***Second***, as suggested by ***Reviewer uzsk (Q3-1)***, we further study the robustness of nuclear norm under the imbalance setting where training and test sets are long-tailed (with power law). We show that nuclear norm is less effective than other methods under strongly long-tailed test sets. However, under moderately imbalanced test sets, nuclear norm is still competitive with other methods.
>
> - ***Third***, we emphasize that considering the prediction dispersity under strong imbalanced test sets is still useful. Specifically, if we have prior knowledge about the long-tailed class distribution, we can expect class predictions to follow it rather than a uniform distribution. In this way, we can more accurately characterize the class-specific prediction dispersity for the task of accuracy estimation. Specifically, we have discussed ***two potential ways to improve the measure of prediction dispersity under severe imbalance***:
>
>   ***[1]*** We tried to rectify the nuclear norm to mainly consider “head” classes based on the imbalanced intensity.  Please refer to ***Q3-3/Reviewer UZsk*** for more details.
>
>   ***[2]*** Another potential way is to improve mutual information maximization by using prior knowledge to explicitly consider the class distribution. Specifically, the first term of MI encourages the predictions to be globally balanced. If we have prior knowledge of the class distribution, we can modify the first term to explicitly make the predictions match the known class distribution: Let $p = \frac{1}{n_t}\sum_{i=1}^{n_t}{P_{i,:}}$ denote the prediction distribution across classes, we change the first term $H(p)$ to $D_{KL}(p||q)$, where $q$ is known or estimated class distribution. Note that when $q$ is uniform distribution, $D_{KL}(p||q)=H(p)+\text{log}(k)$, where $k$ is number of classes.
>
> **Q2: Notation of Section 3.1: $\Delta_k$.**
>
> Thanks. We have fixed this notation and double-checked the paper to eliminate typos.

---

> ### Author Response · Authors · 2022-12-02
> **Looking Forward to Your Feedback**
>
> Dear reviewer JXCX:
>
> We would like to thank you for your constructive comments, which are very helpful and make our paper even stronger!
>
> We have thoroughly discussed and studied long-tail/imbalanced datasets. We show that the nuclear norm is still effective for moderately imbalanced test sets (Q1-1). Furthermore, we emphasize that it is still useful to consider prediction dispersity under strongly unbalanced test sets (Q1-2). Specifically, we discuss two potential ways to improve the measure of prediction dispersity under severe imbalance.
>
> We sincerely hope that our response addresses your concerns. We would appreciate it if you would consider updating your score. We remain available and are committed to taking further suggestions to improve our work.
>
> Kind regrads,
>
> Author of Paper133

---

### Official Review · Reviewer_UZsk · 2022-10-23

**Confidence:** 3
**Correctness:** 3
**Technical Novelty And Significance:** 2
**Empirical Novelty And Significance:** 3
**Recommendation:** 6

**Clarity, Quality, Novelty And Reproducibility:**

The novelty is somewhat limited because the nuclear norm has been used in the context of domain adaptation and the connection to confidence and dispersity was already established. To me, this was not sufficiently clear from the beginning; it was indicated at the end of the introduction and only became sufficiently clear in Section 3.3.

Both the clarity and quality of the paper are good. The analysis is mainly empirical but the experiments are comprehensive. However, I think the paper should provide more results for the case of imbalanced datasets and/or label shifts because this is a common issue in practice.

Reproducibility should not be a problem, as most of the experiments are based on existing models and datasets. However, it could be improved by providing code for all experiments.

**Strength And Weaknesses:**

**Strengths**

The paper is quite clear and easy to read. The goal is stated clearly. The empirical results are in line with the claims from the abstract and introduction.

It's helpful that the paper provides an initial motivation by showing the relationship between dispersity and OOD accuracy.

The association between nuclear norm and OOD accuracy is very strong and consistent across datasets. The used visualizations show the trend very clearly.

The paper is transparent about the limitations of the proposed heuristic and points out directions for future work.


**Weaknesses**

The presented setting for unsupervised accuracy estimation seems to be a special case of the unsupervised domain adaptation setup with only one training domain. It is not clear why the problem is not presented from the more general perspective of domain adaptation. To better assess generalization, the evaluation could consider more evaluation metrics, not only accuracy.

The results suggest that the nuclear norm performs best on datasets with balanced classes. However, on imbalanced test datasets, the predictive value of the nuclear norm starkly decreases, whereas at least one of the competing methods seems to be much more robust. In light of these results, it would be helpful to have more thorough ablations with respect to the performance on imbalanced datasets. For instance: What if both the training and test data were highly imbalanced? How well do the other heuristics (AC, ANE, DoC, MI) perform under imbalance and/or label shift? Can you provide an experiment that demonstrates that the nuclear norm can be adapted to deal with label shift?

The nuclear norm seems to consistently overestimate the OOD accuracy for randomly synthesized datasets (CIFAR-10-$\overline{\text{C}}$-Rand). Is there a particular reason for this? It would be helpful if you would also provide the performance metrics for each subplot in Figure 5 and Figure 7.

From the abstract, it sounds like the nuclear norm and its connection to confidence and dispersity would be a core contribution of the paper. Only at the end of the introduction was it indicated, and in Section 3.3 explained, that the nuclear norm has been used for similar applications (domain adaptation) and that the connection to confidence and dispersity was already established by previous work. I think it would be better to state the contribution more clearly/conservatively in the abstract.


**Questions and comments**

In Table 1, the performance improvements compared to existing methods seem relatively small in many cases. Is there a way to compute standard deviations or other uncertainty estimates to quantify the significance of these results?

At the beginning of Section 3.2, the prediction matrix is defined on the real numbers. Shouldn't it be defined on the interval (0, 1) instead?

In Section 3.2 it is suggested that classifiers that do not generalize well tend to give degenerate predictions by assigning test samples to some specific categories. Are there any insights into *which* categories typically correspond to degenerate predictions? Or is it quasi-random and depends more on the seed than on the actual categories?







**Summary Of The Paper:**

The paper proposes a new heuristic to estimate the accuracy of a given model for OOD test observations without labels. It uses the nuclear norm (i.e., the normalized sum of singular values) of the prediction matrix (i.e., a matrix of softmax outputs for the test data) of a given model. The measure is motivated by results from previous work, which show that the nuclear norm integrates the confidence and dispersity of the model predictions. While the confidence was shown to be predictive for the OOD performance by previous work, the paper demonstrates that the dispersity is also a predictive feature. Empirically, the paper shows a strong correlation between the computed nuclear norm and the OOD classification accuracy on multiple datasets with synthetic and realistic corruptions. The proposed heuristic shows a slightly but consistently stronger association with OOD accuracy compared to existing measures across multiple datasets, models, and types of shifts. However, the nuclear norm performs worse on test datasets with imbalanced classes and seems to overestimate the OOD accuracy on randomly synthesized datasets

**Summary Of The Review:**

For the goal of unsupervised accuracy estimation, the paper proposes a simple heuristic that shows a strong association with OOD accuracy across many models, datasets, and types of shifts. It is a solid paper with a thorough empirical evaluation, but the utility for imbalanced datasets is not sufficiently clear. It could be stated more clearly that the idea is heavily based on existing work, and the connection to the more general setup of domain adaptation could be clearer.

---

> ### Author Response · Authors · 2022-11-16
> **Author Response to Reviewer UZsk (Part IV)**
>
> **Q10: Reproducibility should not be a problem … it could be improved by providing code for all experiments.**
>
> Thanks for this suggestion. Our method can be implemented with one line of Pytorch code: nuclear_score = torch.norm(softmax_outputs, p='nuc'). As illustrated in Section B of Appendix, the datasets and trained models are also publicly available. To facilitate the research, we will release all computed logits of models under each setup, so that the results of all methods listed in the experiment can be easily computed.

---

> ### Author Response · Authors · 2022-11-16
> **Author Response to Reviewer UZsk (Part III)**
>
> **Q4: In Table 1, the performance improvements compared to existing methods seem relatively small in many cases. Is there a way to compute standard deviations or other uncertainty estimates to quantify the significance of these results?**
>
> In the task of unsupervised accuracy estimation, the classifier is given and fixed and the test sets are the same for different methods. Therefore, the results of all methods reported in Table 1 are deterministic and there is no randomness or standard deviation.
>
> We would like to clarify that the improvement over existing methods is significant. As shown in Table 1, existing methods can fail to predict the performance of some classifiers (e.g., ConvNeXt of ImageNet setup and ResNet-101 of CUB setup). In comparison, the nuclear norm is more robust and accurate: it gains the highest correlation strength across different models under all three setups. Moreover, the scatter plots of Figures 2, 3, and 4 also show nuclear norm leads to a stronger correlation than existing methods.
>
> **Q5: It would be helpful if you would also provide the performance metrics for each subplot in Figure 5 and Figure 7.**
>
> Thanks for this suggestion. We have included correlation scores in Figures 5 and 7.
>
> **Q6: To better assess generalization, the evaluation could consider more evaluation metrics, not only accuracy.**
>
> Thanks for this suggestion. We newly use F1-Macro as the evaluation metric, which combines precision and recall to measure the classifier performance. We report the average $R^2$ and $\rho$ under ImageNet setup:
>
> |AC|ANE| ATC | DoC | Nuclear Norm |
> |--|--|--|--|--|
> |0.912 / 0.974| 0.891 / 0.967 | 0.923 / 0.975 | 0.912 / 0.974 | **0.968** / **0.983** |
>
> We observe that Nuclear Norm is still effective under F1-Macro and its average correlation strength (both $R^2$ and $\rho$) is higher than other methods.
>
> **Q7: At the beginning of Section 3.2, the prediction matrix is defined on the real numbers. Shouldn't it be defined on the interval (0, 1) instead?**
>
> Thanks. We have revised the text to clarify that each row of the prediction matrix is the softmax output of a test sample. So its values are defined on the interval (0, 1).
>
> **Q8: The nuclear norm seems to consistently overestimate the OOD accuracy for randomly synthesized datasets (CIFAR-10-C-Rand). Is there a particular reason for this?**
>
> We think there are two reasons. ***First***, CIFAR-10-C-bar-Rand is synthesized from three randomly selected corruption types with random strength. This synthesis is different from the controlled creation manner of CIFAR-10-C. Therefore, the linear trend of CIFAR-10-C is slightly different from CIFAR-10-C-bar-Rand: the linear fit on CIFAR-10-C overestimates the OOD accuracy of CIFAR-10-C-bar-Rand.
> ***Second***, we further observed that, at similar levels of accuracy, the more severely corrupted test set of CIFAR-10-C-bar-Rand tends to have higher prediction dispersity (more classes are involved and predicted) than CIFAR-10-C, and thus has higher nuclear norm. This make them under the linear line.
>
> **Q9: In Section 3.2 it is suggested that classifiers that do not generalize well tend to give degenerate predictions by assigning test samples to some specific categories. Are there any insights into which categories typically correspond to degenerate predictions? Or is it quasi-random and depends more on the seed than on the actual categories?**
>
> Insightful question. Inspired by the question, we further analyzed how the predictions degenerate by examining the confusion matrix. Under CIFAR-10 and CUB setups, we observe a severe prediction degeneration: test samples tend to be only assigned to a few specific classes when the corruption shift increases. Moreover, the degeneration is relatively mild under the ImageNet-1K setup and we did not observe specific categories that dominate all predictions.
>
>   1) Under the CIFAR-10 setup, some pairs of classes are mostly confused (e.g., "cat" and "dog"). When the classifier's accuracy is very low at a high corruption intensity, the predictions are mainly assigned  to "cat", "boat", and "airplane".
>
>   2) Under the CUB-200 setup, when the corruption shift increases, the predicted classes degenerate to a few categories/ species that share common color and shape. For example, bird images from the same Genius “FreGatidae” are mainly assigned to "Scarlet Tanager".
>
>   3) Under the ImageNet setup, we observe that images from the same superclass (e.g., Birds, Vehicles, Devices, and Clothes) are most likely to be confused. When the corruption intensity is very high, some classes have more assigned test samples, leading to lower prediction dispersity. However, there is no specific category that dominates all predictions.

---

> ### Author Response · Authors · 2022-11-16
> **Author Response to Reviewer UZsk (Part II)**
>
> **Q3-1: It would be helpful to have more thorough ablations with respect to the performance on imbalanced datasets. For instance: What if both the training and test data were highly imbalanced?**
>
> Thanks for raising this discussion. During the rebuttal, we conducted the experiment under CIFAR-10 setup. We use the publicly available code provided by (Cao et al., 2019) to create highly long-tailed training sets with small imbalance ratios (m={0.05, 0.1, 0.3}). To create long-tailed test sets of CIFAR-10-C, we use the same sampling strategy to sample the head and tail classes of each test set. We report the correlation strength ($R^2$ / $\rho$) in the following:
>
> (1) Imbalance ratio $m = 0.05$:
> |ANE| AC | DoC | ATC| Nuclear Norm |
> |:---:|:---:|:---:|:---:|:---:|
> |**0.930**/**0.987**|0.920/0.976 |0.904/0.977| 0.920/0.976 |0.801/0.814|
>
> (2) Imbalance ratio $m= 0.1$:
> |ANE| AC | DoC | ATC| Nuclear Norm |
> |:---:|:---:|:---:|:---:|:---:|
> |0.930/0.987| 0.925/**0.983** | **0.945**/**0.983** | 0.925/0.983 | 0.905/0.917
>
> (3) Imbalance ratio $m= 0.3$:
> |ANE| AC | DoC | ATC| Nuclear Norm |
> |:---:|:---:|:---:|:---:|:---:|
> |0.922/0.992| 0.921/0.991 | 0.955/0.992 |0.921/**0.993** | **0.985**/**0.993**
>
> We observe that nuclear norm is less effective than other methods under strongly long-tailed test sets (m<=0.1). However, under moderately imbalanced test sets, the nuclear norm is comparable with other methods.
>
> *​​Kaidi Cao, Colin Wei, Adrien Gaidon, Nikos Arechiga, and Tengyu Ma. Learning imbalanced datasets with label-distribution-aware margin loss. In NuerIPS, 2019*
>
>
> **Q3-2: How well do the other heuristics (AC, ANE, DoC, MI) perform under imbalance and/or label shift?**
>
> We have updated Figure 7 to show the correlation results of all methods under imbalanced test sets. We summarize the correlation strength ($R^2/\rho$) of all methods below:
>
> |ANE| AC | DoC | ATC| Nuclear Norm | MI |
> |:---:|:---:|:---:|:---:|:---:|:---:|
> |0.950/0.980| 0.960/0.982 | 0.956/0.982 | 0.970/0.982 | 0.881/0.918| 0.890/0.925|
>
> We observe that both mutual information maximization (MI) and nuclear norm are less effective than other methods under the imbalanced setting.  We further note that MI and nuclear norm are robust under mild-imbalanced test sets (shown in Figure 7).
>
> **Q3-3: Can you provide an experiment that demonstrates that the nuclear norm can be adapted to deal with label shift?**
>
> Thanks for this comment. We tried to relax the regularization of nuclear norm under imbalanced test sets. Nuclear Norm encourages the predictions to be well-distributed across all classes. For imbalanced test sets, we can relax this regularization on the tail classes. That is, we mainly consider the prediction dispersity of head classes.
>
> To achieve this, we explored one intuitive way to rectify the nuclear norm: we modify the normalization (i.e., upper bound) of the nuclear norm. Specifically, we revise the normalization from $\sqrt{\min(n_t, k) * n_t}$ to $\sqrt{\min(n_t, k_\text{head}) * n_t}$, where $k_\text{head}$ is the number of major classes regularised by nuclear norm. We conducted an experiment under ImageNet setup (k=1000) and empirically set $k_\text{head}$ based on the imbalanced intensity $r_m$ (the ratio between the number of last-10 "tail" classes and the number of top-10 "head" classes):  $k_\text{head} = k - (1 - r_m) *80$. To estimate imbalanced intensity, we use BBSE (lipton
>  etal., 2018) to estimate the class distribution.
>
> In the following Table, we show that our attempt (rectified nuclear norm) can improve nuclear norm. The visualization is shown in Figure 9 of Appendix.
>
> | ATC | Nuclear Norm | Rectified Nuclear Norm
> |:--:|:--:|:--:|
> |0.970 / 0.982| 0.881 / 0.918 | 0.945 / 0.956 |
>
> We would like to mention that it is unlike to achieve perfect modification of nuclear norm in the short run. As such, we would like to view the above experiment as a starting point that inspires more research on the rectification of nuclear norm for strong imbalanced test sets.
>
> *Lipton, Zachary, Yu-Xiang Wang, and Alexander Smola. "Detecting and correcting for label shift with black box predictors." In ICML, 2018*

---

> ### Author Response · Authors · 2022-11-16
> **Author Response to Reviewer UZsk (Part I)**
>
> **Q1: Unsupervised accuracy estimation seems to be a special case of the unsupervised domain adaptation setup with only one training domain. Why the problem is not presented from the more general perspective of domain adaptation?**
>
> We would like to clarify that unsupervised accuracy estimation is not a special case of unsupervised domain adaptation. They are significantly different tasks.
>
> - First, ***the two tasks have different settings and goals***. Unsupervised domain adaptation considers a fixed pair of source-target datasets. Given labeled source data and unlabeled target data, its goal is to learn an adaptive model that generalizes well to the unlabeled target domain. In comparison, unsupervised accuracy estimation considers various target datasets and a trained model. This goal is not to adapt the model to the target data. Instead, its goal is to estimate the performance of the trained and fixed model on various unlabeled test sets.
> - Second, ***the two tasks have different research directions***. Unsupervised domain adaptation works develop domain adaptive algorithms to eliminate domain discrepancy. In contrast, unsupervised accuracy estimation methods typically derive model-based distribution statistics of test sets (e.g., DoC and ATC) for the accuracy estimation.
>
> In summary, domain adaptation aims to improve model accuracy on an unlabeled test set, while accuracy estimation aims to estimate model accuracy on various test sets. Therefore, we follow the standard setting (Deng etal 2021; Garg etal 2022;Yu etal 2022) to study unsupervised accuracy estimation. We have highlighted the difference in Section 3.1.
>
> **Q2-1: The novelty is somewhat limited because the nuclear norm has been used in the context of domain adaptation and the connection to confidence and dispersity was already established.**
>
> We respectfully disagree with this comment. We would like to highlight our novelty and contribution from the following three aspects.
>
> - First, our ***new insight is that prediction dispersity is an important property*** for the task of accuracy estimation. This motivates us to consider both prediction confidence and prediction dispersity. To this end, we use nuclear norm in this work. Furthermore, we show  that mutual information maximization is another feasible way to capture both properties of the prediction matrix.
>
> - Second, nuclear norm is one of the common attributes of a matrix and it has been studied in the tasks of low-rank optimization and domain adaptation. In this work, we ***explore its new application***: estimating classifier accuracy.
>
> - Third, we note that domain adaptation and accuracy estimation have significantly different settings and goals (Please refer to Q1 above). Therefore, the ***nuclear norm is used for different purposes*** in domain adaptation approaches and this work. Specifically, The nuclear norm is used as a regularization in each mini-batch to help the training procedure of the domain alignment. In our work, nuclear norm is used to characterize the prediction matrix on all unlabelled test samples. We show it can well reflect the distribution statistics of test sets and thus is predictive of classifier accuracy.
>
> **Q2-2: I think it would be better to state the contribution more clearly/conservatively in the abstract.**
>
> Thanks for this suggestion. We have revised Abstract to clearly state our contribution. We use the following sentences:
>
> *While recent methods study the prediction confidence, this work newly reports prediction dispersity is another informative cue. Confidence reflects whether the individual prediction is certain; dispersity indicates how the overall predictions are distributed across all categories.
> Our key insight is that a well-performing model should give predictions with high confidence and high dispersity. Specifically, we need to consider both properties so as to make more accurate estimates. To this end, we use the nuclear norm which has been shown to characterize the two properties.*

---

> ### Comment · Reviewer_UZsk · 2022-11-22
> **Response to the authors**
>
> Thank you for the detailed answers and the revision of the manuscript.
>
> In light of the new results, I think it would be helpful to readers if the paper explained the limitations in the abstract more concretely (currently, it only says “we study the limitation of the nuclear norm and discuss potential directions”). Specifically, one could point out that the improved estimation applies to mostly balanced datasets.
>
> I raise my recommendation from 5 to 6.

---

> > ### Author Response · Authors · 2022-11-22
> > **Thank you**
> >
> > Dear Reviewer UZsk,
> >
> > We appreciate your constructive comments and thank you for raising your score.
> >
> > We will use the following sentence in Abstract: *we investigate the limitation of the nuclear norm, study its improved variant under severe class imbalance, and discuss potential directions*.
> >
> > Best Regards,
> >
> > Authors

---

### Official Review · Reviewer_eGpU · 2022-10-30

**Confidence:** 3
**Correctness:** 4
**Technical Novelty And Significance:** 3
**Empirical Novelty And Significance:** 4
**Recommendation:** 8

**Clarity, Quality, Novelty And Reproducibility:**

Clarity: the paper is quite clear and easy to follow. I would recommend moving some intuition for why nuclear norm captures dispersity to the main paper, even though this is from prior work. It seems crucial to appreciating and understanding the proposed method.

Quality: the experiments are exhaustive and claims are substantiated. The paper also addresses limitations of the method appropriately, and hence I rate the paper as high quality.

Originality: the paper takes an existing idea of dispersity (and how it can be approximated by nuclear norm of the prediction matrix) and applies it to the problem of unsupervised accuracy estimation. It is a novel application of an existing idea.

**Strength And Weaknesses:**

Strengths:
(i) The paper studies a very important problem of unsupervised accuracy estimation under distribution shifts.
(ii) The paper considers an interesting idea and is well motivated. The paper is overall well-written and easy to follow as well.
(iii) The paper performs very systematic and thorough experiments on a large number of datasets.

Weaknesses:
(i) The paper can do a better job at synthesizing the results of the experiments, especially because they could reveal interesting insights about deep networks or distribution shifts, beyond just providing a method that improves accuracy estimation. For e.g. why is ObjectNet always off the line---is this because of the label shift?
(ii) The paper is missing some comparisons to recent related work: Baek et al. 2022 (agreement-on-the-line) and projection norm (Yu et al. 2022). Baek et al. is a recent paper, but projection norm was older.

**Summary Of The Paper:**

This paper studies accuracy estimation from unlabeled data, in the presence of distribution shifts. They show that "dispersity" that captures the marginal distribution of the predicted labels correlates well with the accuracy. The paper assumes that the true label distribution is uniform and proposes a measure of dispersity that measures how much predicted label distribution deviates from uniform. This measure builds off of previous works on nuclear norm. Intuitively, this won't work when there is label shift and the paper shows that nuclear norm based estimation does indeed fail under large label shifts but is able to handle moderate label shifts.

**Summary Of The Review:**

This paper presents an interesting and intuitive idea of using dispersity in model predictions for estimating accuracy on unlabeled data from a shifted distribution. The results are compelling and I think this would be an interesting contribution to the ICLR community and inspire follow-up work. The paper could do a better job with exploring more interesting takeaways and insights based on the strong correlation between dispersity and performance. The paper could also benefit from exploration of other metrics (not based on nuclear norm) that also try to incorporate the marginal of the predictions to determine OOD performance.

---

> ### Author Response · Authors · 2022-11-16
> **Author Response to Reviewer eGpU**
>
> **Q1: The paper can do a better job at synthesizing the results of the experiments, especially because they could reveal interesting insights about deep networks or distribution shifts, beyond just providing a method that improves accuracy estimation. For example, why is ObjectNet always off the line---is this because of the label shift?**
>
> Thank you for this insightful suggestion. Apart from the importance of prediction dispersity, we have several interesting observations.
>
> - ***First***, ObjectNet is built in a bias-controlled manner (with controls for rotation, background, and viewpoint), and its classes are moderately imbalanced. We observe that its images are often confidently misclassified, which makes predictions with the high nuclear norm. We believe this is why ObjectNet is always off the line.
>
> - ***Second***, inspired by the suggestion of ***Reviewer uzsk (Q9)***, we observe that when corruption shift increases, test samples tend to be assigned to a few specific categories under both CIFAR-10 and CUB-200 setups. Moreover, we did not observe such severe prediction degeneration under ImageNet-1K setup.
>
> - ***Third***, for all accuracy estimation methods, they can well capture the model performance is high (top-right region of each figure). However, when model accuracy is low (bottom-left), existing methods cannot make reasonable estimations, especially under CIFAR-10 and CUB-200. In contrast, nuclear norm can well handle the low-accuracy region by additionally considering the prediction dispersity. To improve the accuracy estimation, *it would be helpful to further consider the characteristics of predictions when the model performs poorly*.
>
> - ***Fourth***, in Figures 2,3, and 4, we observe that the real-world test sets (e.g., ImageNet-R, CINIC, and CUB-P) scatter around the linear lines fit on synthetic datasets. This indicates that both real-world and synthetic datasets follow a similar linear trend. This gives an interesting hint: *we can use synthetic datasets to simulate and capture the distributions of real-world test sets*.
>
> In the revised paper, we have highlighted the above observations.
>
> **Q2: The paper is missing some comparisons to recent related work: Baek et al. 2022 (agreement-on-the-line) and projection norm (Yu et al. 2022).**
>
> Thanks for this comment. During the rebuttal, we newly included the two suggested methods for comparison. For a fair comparison, we follow the same setting as ALine-D (Baek et al. 2022). We report the correlation results using ResNet18 on CIFAR-10-C in the following Table:
>
> |Correlation|ProjNorm|ALine-D|Nuclear Norm|
> |:--:|:--:|:--:|:--:|
> |$\rho$| 0.980| 0.995 |**0.997** |
> |$R^2$| 0.973 | 0.974 | **0.990** |
>
> We observe that nuclear norm gains stronger correlation than the two methods. It achieves 0.997 and 0.990 in rank correlation ($\rho$) and coefficients of determination ($R^2$), respectively.
>
> Furthermore, we would like to mention that ALine-D (Baek et al. 2022) requires a set of models for accuracy estimation. ProjNorm (Yu et al. 2022) needs to fine-tune a pre-trained network on each OOD test set with pseudo-labels. In contrast, nuclear norm is more efficient: it is computed on a classifier's prediction matrix on each unlabeled test set.
>
> We have included the above comparison in Section B.5 of the Appendix.
>
> **Q3: I recommend moving some intuition for why nuclear norm captures dispersity to the main paper, even though this is from prior work. It seems crucial to appreciating and understanding the proposed method.**
>
> Good suggestion. Following the suggestion, we have updated Section 3.3 to illustrate why the nuclear norm captures dispersity and confidence.
>
> **Q4: The paper could also benefit from exploration of other metrics (not based on nuclear norm) that also try to incorporate the marginal of the predictions to determine OOD performance.**
>
> Good idea. Our new key insight is that prediction dispersity is an important property and thus should be considered together with confidence. Our experiment (Table 2) further shows that mutual information maximization is another feasible way to incorporate the marginal of the predictions. We hope our work can inspire more research on the characterization of predictions for the task of accuracy estimation.

---

> ### Author Response · Authors · 2022-12-02
> **Looking Forward to Your Feedback**
>
> Dear Reviewer eGpU:
>
> We appreciate your positive and detailed comments. We thank you for your valuable and constructive suggestions.
>
> We have summarized four interesting observations along with our key insights into prediction dispersity (Q1). Two recent methods (ALine-D and ProjNorm) are included for comparison and discussion (Q2). We observe that the nuclear norm is still competitive compared with them. We also show the feasibility and effectiveness of another metric mutual information maximization (Q4). This further validates that characterizing prediction dispersity is useful and informative.
>
> We hope our response addresses your original questions. Please let us know if you have any other comments. We are eager to hear your feedback.
>
> Best,
>
> Author of Paper133

---

### Decision · Program_Chairs · 2023-01-20

**Decision:**

Reject

**Justification For Why Not Higher Score:**

The reviewers all noted the limitation with respect to label imbalance and label shift, which seems to be the setting where this would be applicable in the real world.  In addition, the technical contribution is somewhat limited, i.e. nuclear norm of the prediction matrix and reviewers raised connections to novelty.  Thus it seems reasonable to expect some work / innovation addressing the label shift issue.

**Justification For Why Not Lower Score:**

N/A

**Metareview: Summary, Strengths And Weaknesses:**

In this paper, the authors propose a new way to try to predict a model's classification accuracy on out-of-distribution test data without access to the labels.  Traditionally, one might consider model confidence or the entropy of the predictive distribution.  The authors argue that a notion of dispersity would be more effective, and propose measuring dispersity by computing the nuclear norm of the example x class prediction matrix.  The authors show across a variety of datasets and corresponding out-of-distribution test sets that the nuclear norm is more strongly correlated with classification accuracy than other competing metrics.

The overall review scores were quite mixed with 8, 6, 3 and averaged just below borderline.  Overall, the reviewers felt that the paper was well motivated, well written and easy to follow.  Two reviewers found that the experiments were thorough and the results overall were convincing that in general the proposed method outperformed baselines.

One reviewer found the experiments quite compelling while two found the setting where the methods works best somewhat unrealistic, i.e. when there is a uniform label distribution.  Indeed, in many benchmark datasets the classes are balanced but in practice the class distribution is rarely uniform.  In experiments where the label distribution isn't uniform, the proposed nuclear norm method does not outperform existing methods.  The authors are up-front about that limitation.   All reviewers remarked that this was a clear limitation of the work, but differed significantly in their assessment of how much that mattered.

Other weaknesses raised by the reviewers were that baselines were limited and that novelty is somewhat low, particularly given that the nuclear norm has been explored in existing work before related to prediction confidence.  Another commonly listed weakness was missing baselines (Baek et al. 2022, projection norm, and other confidence-based measures).

Overall, all the reviewers listed ways they felt that the paper could be improved.  This along with the just below borderline average indicates that perhaps the paper isn't quite ready yet and could be stronger with some additional effort.

**Summary Of Ac-Reviewer Meeting:**

Note, we had an asynchronous email thread instead of an in-person meeting because the AC and two of the reviewers had covid during different times in the discussion period.  There was no time everyone could be on the call.  Two reviewers had very different scores (8 and 3) and although they clarified their points, they felt strongly about keeping their scores.